# Cdx2 Animal Models Reveal Developmental Origins of Cancers

**DOI:** 10.3390/genes10110928

**Published:** 2019-11-14

**Authors:** Kallayanee Chawengsaksophak

**Affiliations:** Laboratory of Cell Differentiation, Institute of Molecular Genetics of the Czech Academy of Sciences, v.v.i. Vídenská 1083, 4 14220 Prague, Czech Republic; kchaweng@img.cas.cz

**Keywords:** metaplasia, Cdx, cancer, animal models

## Abstract

The *Cdx2* homeobox gene is important in assigning positional identity during the finely orchestrated process of embryogenesis. In adults, regenerative responses to tissues damage can require a replay of these same developmental pathways. Errors in reassigning positional identity during regeneration can cause metaplasias—normal tissue arising in an abnormal location—and this in turn, is a well-recognized cancer risk factor. In animal models, a gain of *Cdx2* function can elicit a posterior shift in tissue identity, modeling intestinal-type metaplasias of the esophagus (Barrett’s esophagus) and stomach. Conversely, loss of *Cdx2* function can elicit an anterior shift in tissue identity, inducing serrated-type lesions expressing gastric markers in the colon. These metaplasias are major risk factors for the later development of esophageal, stomach and colon cancer. Leukemia, another cancer in which *Cdx2* is ectopically expressed, may have mechanistic parallels with epithelial cancers in terms of stress-induced reprogramming. This review will address how animal models have refined our understanding of the role of *Cdx2* in these common human cancers.

## 1. Introduction

During the development of the embryo, the specification of cellular and tissue identity is dictated according to location. This is achieved through a combination of inductive cues and cell-intrinsic genetic factors. Our current understanding of the fundamental molecular mechanisms that underly these processes, referred to as pattern formation, was initially spurred by the study of the fruit fly, *Drosophila melanogaster*, over a century ago [1]. In 1894 William Bateson reported a peculiar mutation in *D. melanogaster*, in which a leg developed in the place of antennae. This he termed “homeosis”, developmental anomalies which cause one body part to develop in the likeness of another. Genetic mutations which cause homeosis are called homeotic mutations.

Many homeotic mutations have been identified in *D. melanogaster*. These include the bithorax mutation, where an extra pair of wings are present instead of a pair of halteres, and the aforementioned *Antennapaedia* mutation, where legs developed in the place of antennae. The gene responsible for the *Antennapaedia* mutation would be identified almost 90 years later [2] and others soon followed. Comparative sequence analyses indicated that several homeotic genes, including the *Antennapaedia* gene, contained a conserved 180 nucleotide sequence—the homeobox [3,4,5]. Although many genes important for pattern formation were found to contain a homeobox sequence, homeotic transformations in *D. melanogaster* were only associated with those genes mapping to a single genetic locus, termed the *HOM-C* locus [6,7,8].

In human, the *HOM-C* homologues are termed the *HOX* clusters. Duplication events during mammalian evolution have produced four separate *HOX* clusters: *HOXA, HOXB, HOXC* and *HOXD* [9]. The expression of genes within both the *HOX* and *HOM-C* clusters are spatio-temporally regulated; those located at the 3’-end are expressed earlier and in more anterior regions, while those located at the 5’-end are expressed later and in more posterior regions [10,11,12,13]. 

Mutations in *HOX* genes do not cause the dramatic anatomical transformations observed in *D. melanogaster*, as mammalian development is much less dependent on segmental structures. Only branchial arches, the hindbrain and somites appear to develop on a truly segmental basis, and here the role of *HOX* genes in controlling development of these structures is well documented [13]. In the mouse, loss-of-function mutations of *Hox* genes often cause anterior homeotic transformations—an anterior transformation being when a segmental unit acquires the characteristics of one more rostral. Anterior transformations of the axial skeleton have been reported for several *Hox* null mutants, including *Hoxa2* [14,15], *Hoxb4* [16], *Hoxc8* [17], *Hoxd3* [18] and *Hoxd13* [19]. 

An additional paralogous *Hox* cluster (*ParaHox*) also exists and, like the *Hox* cluster, also exhibits spatio-temporal co-linearity [20]. Both gene clusters are evolutionarily ancient, splitting from a common ancestral *ProtoHox* cluster prior to the split between Bilaterians and Cnidarians, i.e., before the establishment of body plans with bilateral symmetry [21]. In humans, the *ParaHox* cluster consists of three genes, *GSH, PDX1* and *CDX2*. 

As with many of the *Hox* genes, loss-of-function mutation of *Cdx2* in mice is associated with anterior homeotic transformation of the axial skeleton [22]. Similarly, loss-of-function mutation of the related paralogue *Cdx1* also causes anterior homeotic shifts [23] and these patterning defects become further exacerbated in *Cdx1/Cdx2* compound mutants [24]. Null mutants of the third paralogue, *Cdx4*, do not exhibit skeletal defects, but exacerbate the axial patterning defects of both *Cdx1* and *Cdx2* mutants [25]. These findings illustrate not only functional overlap, but show that their collective activity is required to achieve wild-type levels of functional activity—i.e., their functional overlap does not equate to functional redundancy. As such, any genetic or environmental factors that alter Cdx protein levels can have significant effects on establishing positional identity. This is true not only during embryogenesis, but also following regenerative tissue repair in adult tissues, where the reestablishment of positional identity can be required. Incorrect reprogramming of tissue identity in adult tissues is termed metaplasia and metaplasia is increasingly recognized as a major risk factor for developing cancer. This review will focus on the function of *Cdx2* and, less so, its paralogues, *Cdx1* and *Cdx4*, and how genetically engineered mutations of these genes have provided us with animal models that have spurred our understanding of the important links between metaplasia and cancer.

## 2. Metaplasia is an Important Etiological Factor in Cancer

Metaplasia has long been recognized as a risk factor for cancer development and most often follows a common sequence: an environmental insult will cause tissue damage and, in the course of renewal, this tissue may transdifferentiate into a tissue type inappropriate for its location. An early recognized example is the often observed transition from a normal columnar bronchial epithelium to a squamous epithelium in the lungs of smokers, a metaplastic change that is believed to be the site of origin of lung cancers [26]. In 1985, Jonathan Slack proposed that many of these epithelial metaplasias may be analogous to homeotic transformations [27]. He proposed that epithelial stem cells may sometimes be reprogrammed back to an early ontological state and then, as normal progression proceeds, can acquire a new stable epigenetic state that is phenotypically anteriorized or posteriorized. This hypothesis was bolstered by findings showing an anterior shift in epithelial identity in the focal regions of the large intestine of *Cdx2* mutant mice, occurring concomitant with anterior shifts in the axial skeleton [22,28,29]. Later studies would show that targeted overexpression of *Cdx2* could induce metaplasias in the gut, in which anterior epithelial structures were replaced with posterior structures, i.e., directly analogous to a posterior homeotic shift [30,31,32,33,34] (Table 1, Figure 1). Thus, *Cdx2* insufficiency was associated with shifts in the opposite direction to conditions where *Cdx2* was overexpressed. Nevertheless, in both cases, the shifts were associated with cancer progression pathways. As will be discussed in the following sections, these animal models have been valuable in furthering our understanding of cancers of the esophagus, stomach and colon, as well as its less understood oncogenic role in leukemia. 

Environmental insults can induce reprogramming of the gut epithelium. These metaplasias and metaplasia-like (i.e., serrated polyps in the colon) alterations can confer risk for subsequent development of cancers of the gastrointestinal tract. 

## 3. Esophageal Cancer

### 3.1. Barrett’s Esophagus is a Major Risk Factor for Human Esophageal Adenocarcinoma

Barrett’s esophagus is a metaplastic change of normal squamous esophageal epithelium to an abnormal columnar epithelia with gastric and intestinal features [36,37]. The metaplasia arises as a consequence of the epithelial damage and inflammatory response wrought by chronic acid reflux. Patients diagnosed with Barrett’s esophagus have an approximate 100-fold increased risk of developing esophageal adenocarcinoma [38] and, because this metaplastic transformation appears irreversible, preventative measures have relied on controlling acid reflux, primarily through the use of proton pump inhibitors (PPIs) [39]. A recent clinical trial has shown that patients with existing Barrett’s esophagus can still reduce their risk of developing adenocarcinoma by taking PPIs [40]. 

Not being present in the normal esophagus, *CDX2* expression is a biomarker for Barrett’s esophagus [41,42,43,44]. Moreover, *CDX2* expression can often be found in esophageal squamous epithelia inflamed by acid reflux, suggesting that its expression precedes the metaplastic transformation [41,45]. *CDX2* is a direct transcriptional target of the key inflammatory mediator NF-κB [46]. Thus, it is likely that the onset of *CDX2* expression is due to activated NF-κB, which has also been shown to be present in pre-metaplastic inflamed squamous epithelia [47,48]. *CDX2* expression is maintained if the metaplasia advances to an adenocarcinoma, but expression diminishes as the cancer loses epithelial morphology [49,50].

### 3.2. Animal Models Reveal Functional Roles for Cdx2 in Barrett’s Esophagus

A *keratin 14* promoter was used to force *Cdx2* overexpression to the squamous epithelia of mouse esophagus [31]. This was sufficient to induce metaplastic changes in the esophagus that resembled Barrett’s esophagus, but lacked the intestinal goblet cells that are characteristic of the disease in humans. The transition from squamous to columnar epithelia was also associated with a decrease in barrier function, leading to the hypothesis that the transformed epithelia could, in turn, be more sensitive to reflux esophagitis, reinforcing the transition [31]. A similar model was generated in zebrafish, using a *keratin 5* promoter, to drive expression of *cdx1b* [30]. Like the transgenic mouse model, the zebrafish model exhibited similar metaplastic changes in the esophagus but, once again, without any appearance of goblet cells. More recently, a discrete transitional columnar epithelium was found to exist at the junction of the stomach and esophagus [34] and may represent the true source of Barrett’s esophagus. *Keratin 7* was identified as a specific marker of this transitional epithelium and, by employing a *keratin 7* promoter to confine inducible *Cdx2* overexpression to this cell-type, a metaplasia that included goblet cells was observed [34]. Currently, this compound transgenic model (*Krt7rtTA*; *otet-CDX2-T2A-mCherry*) represents the best animal model for replicating Barrett’s esophagus.

## 4. Stomach Cancer

### 4.1. CDX2 and the Metaplastic Origins of Human Stomach Cancer

The current model for human gastric carcinogenesis, proposed in 1992 [51], follows a similar course to the model for esophageal carcinogenesis discussed earlier, in that an initial pro-inflammatory stimulus will lead to inflammation (gastritis), followed by intestinal metaplasia, and, ultimately, to adenocarcinoma. In the stomach, the major environmental stimulus initiating this pathway, and thereby conferring the risk of cancer development, is chronic infection with *Helicobacter pylori* [52]. 

Two major types of metaplastic lineages have been identified adjacent to cancers of the stomach: an intestinal-type metaplasia with the characteristic presence of goblet cells [53] and a spasmolytic polypeptide-expressing metaplasia (SPEM) which expresses trefoil factor 2 (TFF2), then designated as spasmolytic polypeptide [54]. SPEM exhibits similarities to glands of the antrum (the caudal-most region of the stomach) [55,56]. SPEM may represent a reparative response to acute gastritis, while goblet-cell intestinal metaplasia may require a chronic inflammatory environment.

*CDX2* expression is detected in gastric intestinal metaplasia but not in normal gastric mucosa [57,58,59]. *CDX2* could also be detected in chronic gastritis without evidence of metaplasia, suggesting that the onset of *CDX2* expression preceded the metaplastic change [60]. As the metaplasias progresses to carcinoma, *CDX2* levels are often reduced [58].

### 4.2. Animal Models Reveal Functional Roles for Cdx2 in Stomach Cancer

*Cdx2* overexpression in the gastric mucosa of transgenic mice using the parietal cell-specific *H^+^/K^+^-ATPase subunit b* promoter resulted in gastric intestinal-type metaplasia that spontaneously developed into adenocarcinomas [61,62]. Another line, employing the *Foxa3* promoter to drive *Cdx2* overexpression, also exhibited intestinal-type metaplasia but progression to adenocarcinoma was not observed [33]. The significant overlap in phenotype is somewhat surprising, as *Foxa3* is expressed during embryonic development while the *H^+^/K^+^-ATPase* promoter is only active postnatally in the acid-producing parietal cells. It is possible that the phenotype may be indirectly influenced by parietal cell loss, as gastrin knockout mice, which exhibit impairment of stomach acid production, also exhibit intestinal-type metaplasia that eventually progresses to stomach cancer [63,64]. 

Modelling the action of the major risk factor for stomach cancer, *H. pylori* infection, has proven more difficult in mice, as this is heavily mouse strain-dependent [65] and produce SPEM rather than intestinal-type metaplasia [66]. For replicating the human disease, the Mongolian gerbil has been superior, recapitulating upon *H. pylori* infection the progression from gastritis to intestinal metaplasia and, eventually, to gastric cancer [67,68,69].

## 5. Colon Cancer

### 5.1. CDX2 is a Tumour Suppressor in Human Colorectal Cancer

Most colorectal cancers arise from an epithelial-derived adenomatous precursor lesion that, with further mutations in oncogenes and tumor suppressor genes, can clonally progress to carcinoma [70]. This adenoma–carcinoma pathway is most often initiated by activating mutations of the WNT pathway [71]. An alternative pathway, broadly termed the ‘serrated pathway’, maintains epithelial gland morphology and mucin production in the benign precursor lesions. It has been estimated that 20%–30% of colorectal cancers arise by this alternative pathway, although classification can be difficult as the cancer progresses and loses its serrated morphology [72,73]. This pathway is most often associated with activating mutations in the *BRAF* oncogene [72,74], and is considered to follow a more aggressive course than the conventional pathway [75]. Loss of *CDX2* expression has recently emerged as a biomarker for colon cancers arising via the serrated pathway, often coinciding with activated *BRAF* mutations [76,77]. It also has been identified as a prognostic marker in stage II colon cancer, where it was suggested that patients with CDX-negative cancers would benefit from adjuvant chemotherapy, rather than the common practice of treating all stage II patients with surgery alone [78].

Are serrated pathway cancers derived from metaplastic changes in the colonic epithelium? Suggestive of an anteriorization of epithelial identity is the fact that these cancers often express gastric epithelial markers, including mucin 2 (MUC2), MUC5AC, MUC6 and annexin A10 (ANXA10) [79,80,81]. Perhaps more compelling is that loss of *CDX2* expression is associated with a gain in *PDX1* expression, the *ParaHox* gene responsible for patterning the midgut [77]. 

### 5.2. Animal Models Reveal Functional Roles for Cdx2 in Colorectal Cancer

A possible role for *Cdx2* in colon cancer was initially suggested based on the knockout phenotype in mice; heterozygous mice had numerous tumors, although they never spontaneously advanced to carcinoma [22,29]. Upon closer examination, these tumors consisted of small foci of histologically normal forestomach epithelia that were surrounded in successive order by cardia, corpus, antrum and small intestine epithelia [82]. This observation was ascribed as a heterotopia, analogous to a metaplastic transformation, only with its origins occurring during embryological development instead of as a consequence of mucosal injury and repair. To model the latter, a *Cre-ERT* transgene under the *Cyp1a1* promoter, was used to achieve mosaic inactivation of a *Cdx2*^fl/fl^ allele, thus allowing the study of *Cdx2* deficient lesions in the context of wild-type mucosa [83]. The *Cdx2* deficient lesions were found to express a number of gastric genes but did not form normal gastric mucosa, presumably because of incompatible mesenchymal signaling [83]. Under current classifications, these lesions could be interpreted as “serrated”. Could they therefore be susceptible to transformation via activating mutations of *BRAF*?

Mouse models combining *Cdx2* inactivation and oncogenic *BRAF* (*BRAF*^V600E^) activation were recently described and indeed, this led to invasive carcinogenesis along the serrated pathway [77,84]. Tamoxifen-regulated Cre protein (CreERT2) was used to inducibly inactivate *loxP*-containing alleles of *Cdx2* (*Cdx2*^fl/fl^) or to inducibly activate an oncogenic *BRAF* allele (*BRAF*^V600E^) in the adult intestinal epithelium. Mutation of either allele individually had little to no effect on median survival; however, their combined mutation resulted in progression to carcinoma. Immunohistochemical analyses of tumors revealed ectopic expression of typical serrated pathway markers such as annexin A10 and mucin 5AC [77].

Mouse models have also provided information that loss of *Cdx2* expression can influence not only the serrated pathway, but also the classical adenoma–carcinoma pathway. The classical pathway is associated with activating mutations of the Wnt signaling pathway, most predominantly through loss of function of the Wnt-signaling inhibitor, Apc [71]. Mutations in the human *APC* gene are causative for the cancer syndrome Familial Adenomatous Polyposis (FAP), as well as for sporadic cancers arising predominantly in the distal colon [85]. FAP can be modeled in mice carrying mutations of the *Apc* gene, including the truncated mutant *Apc*^∆716^ [86], but tumors in mice arise predominantly in the small intestine. When the *Apc*^∆716^ mutant allele is combined with the *Cdx2*^+/−^ heterozygous mutation, there is a large increase in the number of adenomatous polyps in the distal colon, more closely reflecting the tumor distribution in human FAP [87].

More recently, it was reported that the tumor-promoting effect of *Cdx2* deficiency on the classical adenoma–carcinoma pathway may be non-cell autonomous [35]. This discovery was enabled by a complex mouse model, where mosaic inactivation of a *Cdx2*^fl/fl^ allele was combined with a mutant *Apc*^+/Δ14^ allele to drive adenoma formation and a conditionally activated fluorescent reporter allele (*tdTomato*) to trace cells that underwent Cre-mediated recombination. As expected, adenomas contained high levels of nuclear β-catenin, a measure of hyperactive Wnt signaling arising due to the loss of heterozygosity of the *Apc* tumor suppressor allele. However, these cells were never red (Cdx2 negative). The *Cdx2*-negative cells were not contributing to the adenoma, but instead created an environment that promoted neoplasia of *Cdx2*-positive cells—i.e., *Cdx2* was acting as a “non-cell-autonomous tumor suppressor” [35].

Previous studies had shown that *Cdx2*^+/−^ mice are more susceptible to DSS-induced colitis [88]. Perhaps the permissive environment is pro-inflammatory. Indeed, NF-κB, a key mediator for inflammatory responses, was activated only in the *Cdx2*-positive cells that were adjacent to the regions of incomplete metaplasia [35]. These activated cells also expressed high levels of nitric oxide synthase (iNOS), indicating that these cells were under increased nitrosative and oxidative stress and therefore more susceptible to DNA damage. Supporting this hypothesis, treatment with the iNOS inhibitor aminoguanidine reduced the tumor load in mice carrying a mutant *Cdx2* allele (*Apc*^+/Δ14^; *Cdx2*^+/−^), while having no effect on mice with only wild-type *Cdx2* alleles (*Apc*^+/Δ14^; *Cdx2*^+/+^) [35]. 

## 6. Leukemia

### 6.1. CDX2 is a Proto-Oncogene in Human AML

A possible role for *CDX2* in human acute myeloid leukemia (AML) was first suggested following the identification of a novel chromosomal rearrangement, t (12; 13)(p13;q12), in a patient with AML. The rearrangement yielded an ets variant gene *6–CDX2* (*ETV6–CDX2*) fusion protein [89] and, as ETV6 is an important regulator of HSC survival and is frequently affected by translocations [90,91], it was thought that this fusion may be oncogenic. However, when the fusion protein was transduced into murine hematopoietic progenitors, it caused only minor myeloproliferation, and not transplantable AML [92]. It is now accepted that it was the full-length CDX2 protein, driven from an alternative *ETV6* promoter, that was leukemogenic. Indeed, transduction of full-length CDX2 into murine hematopoietic progenitors does result in transplantable lethal AML [92].

Up to 89% of AML cases, and up to 81% of acute lymphoblastic leukemia (ALL) cases, express *CDX2* [93,94,95,96] and at least for ALL, *CDX2* expression levels were directly associated with the aggressiveness of the disease [93,95]. Thus *CDX2* is one of the most frequently expressed proto-oncogenes in human leukemia. 

Known downstream targets of *CDX2*, namely the *HOX* genes, had also been identified as proto-oncogenes in AML [97,98]. Forced expression of *Hoxa9* or *Hoxa10* are also capable of inducing rapid AML in mice [99,100], and aberrant expression of human *HOX* genes, including *HOXA9*, correlates with clinical measures of disease burden [101,102,103,104]. 

During hematopoiesis, *HOX* genes of the A and B cluster are highly expressed in normal murine and human hematopoietic stem and committed progenitor cells, and become silenced during the course of normal differentiation [105,106]. Bone marrow from *Hoxa9* deficient mice has a profound deficiency in the number of hematopoietic stem cells and progenitors [107,108]. On the other hand, *CDX2* is not detected in hematopoietic stem or progenitor cells from healthy subjects, neither in human nor in mouse [109]. Also, there were no significant effects on hematopoiesis in knockout mouse models of any of the *CDX* genes [22,25,109,110]. Thus, although *CDX2* and the *HOX* genes have similar roles in leukemogenesis, the similarities are not readily apparent in regards to the process of normal hematopoiesis. A true functional role would only be revealed as a result of important scientific discoveries in zebrafish.

### 6.2. Cdx Genes are Required for Normal Hematopoiesis in Zebrafish

The first indication that *Cdx* genes may have a functional role in hematopoiesis came from studies in zebrafish, when the causative mutation underlying the autosomal recessive mutation *kugelig* (*kgg*) was identified in the *cdx4* gene [111]. Homozygous *kgg* embryos die early in development (day 5 to 10 post fertilization) with severe tail defects and a prominent reduction in hemoglobin-staining erythroid cells. This phenotype was consistent with the expression pattern of *cdx4*, which became restricted to the posterior end of the embryo during early somitogenesis, prior to the emergence of the posterior blood islands. Furthermore, the in vivo injection of *cdx4* mRNA was able to rescue hematopoiesis in these *kgg* mutants [111].

Zebrafish contain a duplication of the *Cdx1* gene (*cdx1a* and *cdx1b*), while lacking a *Cdx2* orthologue. Therefore, although they contain the same number of *Cdx* genes as in mammals, they lack the prototypical *ParaHox* cluster. Nevertheless, like in mammals, the zebrafish cohort of *Cdx* genes does exhibit some degree of functional redundancy. Indeed, morpholino-mediated knockdown of *cdx1a* in *kgg* mutant fish exacerbates the phenotype, causing a complete failure to specify blood [112].

The hematopoiesis defect in *cdx4* mutant zebrafish is reminiscent of anterior homeotic transformation of the axial skeleton observed in mouse loss of function mutants [22,110], as there appeared to be a posterior shift in the boundary between anteriorly localized hemangioblasts, fated to develop into endothelial cells and the posteriorly localized hemangioblasts, fated to develop both blood and endothelial cells [111]. Both populations are labelled with scl (tal1), which coexpresses with cdx4 in the posterior blood islands [111]. Even though scl overexpression is able to expand hematopoietic cell numbers when overexpressed in wild-type zebrafish embryos [113], it was incapable of rescuing hematopoiesis in *cdx4* mutant *kgg* embryos [111]. Thus, the hematopoietic defect did not seem to be due to an overt lack in the number of scl+ hemangioblast progenitors, but rather a failure to pattern these progenitors to favor differentiation towards the erythrocyte lineage. 

An evolutionarily conserved role of *Cdx* genes in regulating the expression of *Hox* genes [110,114,115] appears to underly the failure to pattern scl+ hemangioblasts in zebrafish. Indeed, *kgg* mutants exhibit large alterations in *hox* expression patterns, which can be restored upon ectopic *cdx4* expression [111]. Overexpression of several of the most downregulated *hox* genes (*hoxb6b*, *hoxb7a* and *hoxa9a*) successfully rescues hematopoiesis in *kgg* mutants [111], and overexpression of *hoxa9a* (but not *hoxb7a*) rescues the complete hematopoietic failure observed upon combined *cdx1a* and *cdx4* deficiency [112].

### 6.3. Cdx Genes Control Mammalian Hematopoiesis

The implications from these studies in zebrafish were that a functional role for mammalian *CDX* genes may be masked by functional redundancies and that the *CDX* genes may exert their function not at the level of hematopoetic stem cells, but by pre-patterning their early mesodermal progenitors during embryogenesis. This possibility could be simply assessed by the in vitro differentiation of embryonic stem cell lines, since differentiation towards hematopoietic lineages involves a transition through a hemangioblast intermediate.

In vitro differentiation of single *Cdx* gene deficient murine embryonic stem cell lines resulted in only minor reductions in the numbers of multipotential blood progenitor colonies [116]. Knockdown of either *Cdx1* or *Cdx2* by RNA interference in a *Cdx4*-deficient background resulted in more severe reductions, while combined knockdown of both *Cdx1* and *Cdx2* in the *Cdx4*-deficient background resulted in an almost complete failure of hematopoiesis [116]. In embryos where *Cdx2* was conditionally inactivated in a *Cdx1*^−/−^ background, there were defects in primitive hematopoiesis as well as yolk sac vascularization [117]. Thus a previously unrecognized role for *Cdx* genes in hematopoiesis was made evident when all genes in the family were inactivated. 

The role of *Cdx* genes in pre-patterning early presomitic mesodermal progenitors, which will later give rise to hematopoietic lineages, can first of all be inferred by their ability to pattern the somitic mesoderm, resulting in the prototypical anterior homeotic transformation of the vertebrae [22,110]. Also, upon in vitro differentiation of human and mouse embryonic stem cells, *Cdx* gene expression peaks at the same time as hemangioblasts are specified and, if inducibly overexpressed during this time window, strongly enhances the production of hematopoietic progenitors [118,119,120]. The effect on hemangioblast production is likely the result of both a decreased amount of posterior unsegmented mesoderm [121] and an anterior shift in patterning the mesoderm [122]. In zebrafish, the *tbx5a*-expressing anterior cardiogenic mesoderm was expanded in *cdx1a/4* mutants [122]. Similarly, in mice, ectopic *Tbx5* expression was observed in the yolk sac of *Cdx1/2* compound conditional null mutants at the expense of hematopoietic markers [123]. Current evidence supports a mechanism of action for *Cdx* genes in which they stably repress cardiac loci in early Mesp1+ mesoderm by directly recruiting the SWI/SNF epigenetic silencing complex [123]. Thus, the expression of *Cdx* biases these progenitors to hematopoietic lineages at the expense of cardiac lineages.

## 7. Summary

Metaplasias, long recognized as a cancer risk factor, have been suggested to be analogous to developmental homeosis, where normal tissues develop in an abnormal location [27]. Homeobox genes, including *Cdx2*, are important factors in conferring positional identity to developing tissues, whether during embryogenesis or during the regenerative process following tissue injury. Animal models in which *Cdx2* overexpression is targeted to the esophagus show Barrett’s metaplasia (Barrett’s esophagus), characterized by the presence of intestinal-type epithelia in place of normal squamous epithelia. [30,31,34]. Similarly, targeted overexpression of *Cdx2* in the stomach also causes metaplasia, with a posteriorization of epithelial identity [32,33]. These tissue alterations model pre-neoplastic metaplasias that are common in humans. Conversely, loss of *Cdx2* in the colon causes metaplasia-like alterations, in which epithelia are misallocated towards an identity characteristic of more anterior structures [22,28,29], and this has provided important insights into understanding the progression of human serrated-type colonic tumors. 

It is easy to recognize epithelial metaplasias, as any change in the reacquiring of positional identity in an epithelial stem cell will be conferred as a change in phenotype in its regionally constrained cell progeny. However, this is not the case for another cancer in which *Cdx2* is ectopically expressed—leukemia. Nevertheless, it is possible that the same sequence of events is occurring in leukemia—chronic inflammatory damage triggering a regenerative response, which results in the acquisition of a more “posteriorized” epigenetic state. 

While the importance of *CDX2* in human cancer pathology is indisputable, its functional role has been more difficult to define. It has been designated, somewhat contradictorily, as both as an oncogene and a tumor suppressor. But, unlike prototypical oncogenes and tumor suppressor genes, there is no strong statistical evidence for cancer-associated mutations or loss of heterozygosity. The issue is that these terms describe cell-intrinsic functions, while the core function of *CDX2*, as a designator of positional identity, is, by definition, relativistic. Therefore, a true understanding of its role in cancer progression must be context-dependent. Indeed, the conceptual advances in this field, as discussed in this review, have been driven almost exclusively by the judicious use of animal models.

## Figures and Tables

**Figure 1 genes-10-00928-f001:**
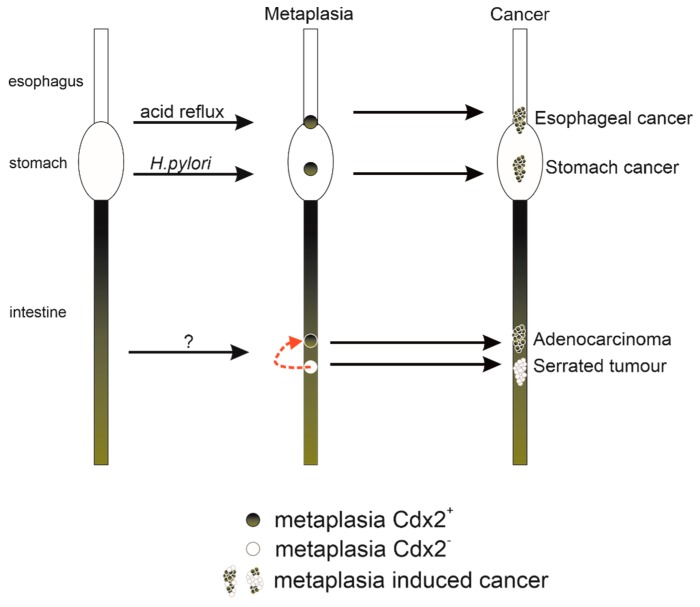
Simplified schematic diagram depicting how metaplasia caused by the alteration of *Cdx2* expression can progress to cancer.

**Table 1 genes-10-00928-t001:** Animal cancer models generated through the genetic manipulation of Cdx genes.

Mutation	Phenotype	Reference
*Cdx2* ^KO^	Homozygotes: preimplantation lethality Heterozygotes: anterior homeotic shift of vertebrae, nondysplastic colonic tumors often containing metaplastic/heterotopic foci with gastric features	[22,28,29]
*Cdx2*^CKO^; *Apc*^+/Δ14^	Mixed tumors with adenomatous and serrated features	[35]
*Tg*(*Foxa3–Cdx2*)	Metaplasia in stomach	[33]
*Tg*(*Atp4a–Cdx2*)	Metaplasia in stomach	[32]
*Tg*(*K14–Cdx2*)	Non-intestinal type metaplasia in esophagus	[31]
*Tg*(*Krt7rtTA*); Tg(*otet-Cdx2*)	Intestinal type metaplasia in esophagus	[34]
*Tg*(*krt5:cdx1b–EGFP*) ^1^	Metaplasia in esophagus	[30]

^1^ Transgenic zebrafish model.

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
