# Peer review of "Cdx2 Animal Models Reveal Developmental Origins of Cancers"

_genes, 2019, doi:10.3390/genes10110928_

Round 1

Reviewer 1 Report

This is an excellent review of the potential roles of CDX2 in cancer with a comprehensive look at the underlying models and what they tell us.

It would be very helpful if the review added a more concise accounting of CDX2 in human cancer. It is dispersed throughout the article but a summary of CDX2 data say from the TCGA analyses in text and table would strengthen the review.

Author Response

Dear Reviewer#1

I thank you for your time. I provide here a response to your points:

Point 1: It would be very helpful if the review added a more concise accounting of CDX2 in human cancer. It is dispersed throughout the article but a summary of CDX2 data say from the TCGA analyses in text and table would strengthen the review.

Response 1: I agree that such a table would be helpful, but only if CDX2 mutation was significantly represented in, for example, the TCGA database. This is actually not the case. I have added a concluding paragraph that emphasizes the point, including the phrase ".. there is no strong statistical evidence for cancer-associated mutations or loss of heterozygosity." My main aim was to convey the role of CDX2 in preneoplastic transitions (metaplasias) which alter cancer risk.

Point 2: Moderate English changes required.

Response 2: I sent the manuscript to two colleagues, Prof. Emeritus Felix Beck and Dr. Trevor Epp for comment and the revised review article has incorporated their suggested edits. 

Reviewer 2 Report

This manuscript attempts to summarize Cdx2-related transgenic animal models. The authors clearly described the functions of Cdx2 family proteins. Some suggestions are provided here:

The role of Cdx2 in human cancer ?? Whether Cdx2 is critical in human cancer development ?? Whether Cdx mutations were identified in human cancer ?? A Table to summarize the correlation of Cdx2 with various types of human cancer will benefit the quality of manuscript.  The organization of Cdx2-transgenic animal models in a Table, including zebrafish, mouse, drosophila.....; The current Table 1 is not complete. The writing style, grammar and spelling should be carefully checked. English editing by naive scientist is suggested. 

Author Response

Dear Reviewer #2

Thank you for sharing your valuable time in assessing this review manuscript. I respond to your points forthwith:

Point 1: The role of Cdx2 in human cancer?? Whether Cdx2 is critical in human cancer development??

Response 1: The four sections, each dealing with a different cancer, contains a first subsection, focused entirely on the current state of knowledge regarding the human condition and the role of CDX2 therewith. It is only in the second subsections, that the respective animal models are introduced. 1/3 of the text is devoted to discussion of the human cancer.

Point 2: Whether Cdx mutations were identified in human cancer?? A Table to summarize the correlation of Cdx2 with various types of human cancer will benefit the quality of the manuscript.

Response 2: I have added a concluding paragraph to the review which makes the main thesis more explicit. It includes the statement that "..there is no strong statistical evidence for cancer-associated mutations". It is rather its role in metaplasia, a response to inflammatory tissue damage, that changes the cancer risk profile - a preneoplastic event.

Point 3: The organization of Cdx2-transgenic animal models in a Table, including zebrafish, mouse, drosophila.....; The current Table 1 is not complete.

Response 3: This was my original goal as well, however the number of transgenic models is quite large and complex (incl. tetraploid and diploid chimaeras, compound transgenic) and mostly assessed developmental mechanisms (e.g. axial skeleton development). I saw that it was introducing unneeded complexity that interfered with the main thesis of the review and the focus of this Special Issue. This is the reason that the table, as titled, lists only the subset of Cdx animal models that are cancer-related.

Point 4: The writing style, grammar and spelling should be carefully checked. English editing by a naive scientist is suggested.

Response 4: The manuscript was sent to two colleagues: Prof. Emeritus Felix Beck and Dr. Trevor Epp, both native English scientists, who have provided me with their suggestions. The manuscript now includes their suggestions and their input is reflected in the current Acknowledgements.

Round 2

Reviewer 2 Report

Authors responded to my concerns point by point.

I have no further questions